# A SNaPshot Assay for Determination of the Mannose-Binding Lectin Gene Variants and an Algorithm for Calculation of Haplogenotype Combinations

**DOI:** 10.3390/diagnostics11020301

**Published:** 2021-02-13

**Authors:** Jana Mrazkova, Petr Sistek, Jan Lochman, Lydie Izakovicova Holla, Zdenek Danek, Petra Borilova Linhartova

**Affiliations:** 1Department of Pathophysiology, Faculty of Medicine, Masaryk University, Kamenice 753/5, 625 00 Brno, Czech Republic; jmrazkova.pracovni@gmail.com (J.M.); Jlochman@seznam.cz (J.L.); holla@med.muni.cz (L.I.H.); danek002@gmail.com (Z.D.); 2Department of Stomatology, Institution Shared with St. Anne’s Faculty Hospital and Faculty of Medicine, Masaryk University, Pekarska 664/53, 656 91 Brno, Czech Republic; 3Secondary Technical School Brno, Purkynova, a Contributory Organization, Purkynova 97, 612 00 Brno, Czech Republic; sistek.petr@email.cz; 4Department of Biochemistry, Faculty of Science, Masaryk University, Kotlarska 267/2, 611 37 Brno, Czech Republic; 5Clinic of Maxillofacial Surgery, Institution Shared with the University Hospital Brno, Faculty of Medicine, Masaryk University, Jihlavska 20, 62500 Brno, Czech Republic; 6Institute of Medical Genetics and Genomics, Faculty of Medicine, Masaryk University, Kamenice 753/5, 625 00 Brno, Czech Republic; 7Department of Molecular Pharmacy, Faculty of Pharmacy, Masaryk University, Palackeho tr. 1946/1, 612 00 Brno, Czech Republic

**Keywords:** mannose-binding lectin, mannose-binding lectin gene (*MBL2*), SNaPshot assay, single nucleotide polymorphism, genotyping, Haplogenotype Calculator, haplogenotypes

## Abstract

Mannose-binding lectin (MBL) deficiency caused by the variability in the *MBL2* gene is responsible for the susceptibility to and severity of various infectious and autoimmune diseases. A combination of six single nucleotide polymorphisms (SNPs) has a major impact on MBL levels in circulation. The aim of this study is to design and validate a sensitive and economical method for determining *MBL2* haplogenotypes. The SNaPshot assay is designed and optimized to genotype six SNPs (rs1800451, rs1800450, rs5030737, rs7095891, rs7096206, rs11003125) and is validated by comparing results with Sanger sequencing. Additionally, an algorithm for online calculation of haplogenotype combinations from the determined genotypes is developed. Three hundred and twenty-eight DNA samples from healthy individuals from the Czech population are genotyped. Minor allele frequencies (MAFs) in the Czech population are in accordance with those present in the European population. The SNaPshot assay for *MBL2* genotyping is a high-throughput, cost-effective technique that can be used in further genetic-association studies or in clinical practice. Moreover, a freely available online application for the calculation of haplogenotypes from SNPs is developed within the scope of this project.

## 1. Introduction

Mannose-binding lectin (MBL, also known as mannose-binding protein, mannan-binding protein/lectin, Collectin 1, MBP1, or mannose-binding protein C) is a crucial part of innate immunity. It is classified as an acute-phase protein because its plasma level increases after the onset of inflammation. MBL belongs to a group of soluble pattern-recognition receptors that can recognize pathogen-associated molecular patterns and damage-associated molecular patterns released from activated cells, cells undergoing apoptosis or necrosis, or host cells with altered surface glycosylation patterns related to cancer phenotypes [1,2,3,4,5]. MBL trimers and tetramers are the forms most common in blood circulation due to their high effectiveness in complement activation after interaction with MBL-associated serine proteases (MASP 1, −2, −3) [6,7,8,9]. Furthermore, MBL works as an opsonin and enables phagocytosis independently on the complement [1,4].

The concentration of the functional MBL in blood circulation is affected by the *MBL2* gene variant located on chromosome 10q11.2-21. Alleles B (rs1800450, NM_000242.2:c.161G>A), C (rs1800451, NM_000242.2:c.170G>A) or D (rs5030737, NM_000242.2:c.154C>T), possessing single nucleotide polymorphisms (SNPs) in exon 1, cause amino acid substitution of Gly54Asp, Gly57Glu, or Arg52Cys, respectively. These variants lead to the production of a structurally impaired protein. As a result of the substitutions higher MBL oligomers disintegrate more rapidly into lower oligomers, which are no longer capable of activating the complement system. These alleles are summarily called alleles O, whereas the allele without substitution (normal allele) is called allele A. Alleles H/L (rs11003125) and X/Y (rs7096206) carrying SNPs localized in the promoter region (NM_000242.2:c.-619C>G and NM_000242.2:c.-290C>G, respectively), and allele P/Q (rs7095891) with the SNP in the 5’ untranslated region (5’UTR) of exon 1 (NM_000242.2:c.-66C>T), alter *MBL2* transcription and translation, directly affecting the MBL plasma level [8,10].

Due to strong linkage disequilibrium (LD) among the six SNPs and variation of these SNPs’ frequencies among ethnic groups, only seven MBL secretor haplotypes (allele combinations) are relatively common. HYPA and LYQA haplotypes are associated with high MBL serum levels, LYPA with intermediate levels, and LXPA with low MBL production. Defect haplotypes LYPB, LYQC and HYPD are responsible for expressing undetectable amounts of MBL [10,11]. Nevertheless, the MBL level in circulation depends on a combination of two haplotypes; for example, the HYPA/LXPA haplogenotype is responsible for medium MBL levels while LYPB, LYQC or HYPD combined with LYPA leads to low MBL levels in the serum [12]. MBL deficiency (a common human immunodeficiency, 5–30% of the population is affected) has been associated with increased susceptibility toward infectious diseases and autoimmune diseases [13]. Therefore, determining genotypes of all six SNPs and the resulting haplogenotypes is necessary for predicting the MBL phenotype.

The SNaPshot assay is a minisequencing method widely used for simultaneous detection of polymorphisms in individual genes [14,15] or the whole genome [16]. The assay is sensitive and straightforward, and the results can be automatically evaluated using software. Therefore, this method is very well suited for multiple SNP genotyping in the *MBL2* gene.

The aim of this work is to develop a low cost, accurate, and reliable method for genotyping six functional SNPs in the *MBL2* gene that could be utilized both in research and clinical practice, and to use this method for the genotyping of 328 samples of healthy individuals from the Czech population. Furthermore, we develop a custom-made online application for the calculation of haplogenotypes in this study.

## 2. Materials and Methods

### 2.1. Study Subjects and Sampling

To optimize the originally designed SNaPshot assay, subjects meeting the inclusion criteria were selected from the pool of patients of the Institutions shared with St. Anne’s Faculty Hospital and Faculty of Medicine, Masaryk University, Brno, Czech Republic. Inclusion criteria for this study were: adult (aged 18 years and higher), Czech nationality, systemic health, and willingness to participate. Exclusion criteria comprised the inability to give consent, blood relationship to another participant, history of systemic diseases (such as diabetes mellitus, coronary artery diseases, malignancies, immunodeficiency disorders), immunosuppression attributable to medication, or concurrent illness.

Three hundred and twenty-eight individuals with good systemic health (mean age ± standard deviation of 34.0 ± 13.5 years, 192 women and 136 men) were enrolled in this study; for demographic data, see Table 1. Peripheral blood was collected from each subject in a tube containing 0.5 M ethylenediaminetetraacetic acid (EDTA) and stored at −20 °C within 30 min of collection.

### 2.2. DNA Isolation

Genomic DNA was isolated from peripheral blood leukocytes using proteinase K digestion of cells and isopropanol-phenol-chloroform DNA extraction according to the standard protocol [17]. DNA concentration and purity were determined using a spectrophotometer NanoDrop 1000 (Thermo Fisher Scientific, Waltham, MA, USA) and all DNA samples were diluted to working concentrations of 50 ng/μL.

### 2.3. SNaPshot Assay

The SNaPshot assay was applied for simultaneous determination of six single nucleotide polymorphisms (SNPs) in the *MBL2* gene (rs11003125, rs7096206, rs7095891, rs1800450, rs1800451, rs5030737). The part of the *MBL2* sequence carrying all SNPs of interest was determined using commercial Sanger sequencing (Eurofins Genomics Germany GmbH, Ebersberg, Germany) in 8 samples randomly chosen from the sample pool. These samples served as controls for the optimization of the SNaPshot assay.

#### 2.3.1. Primer Design

Forward and reverse primers, 5’-CTGTAAACAGATTCCCCCAGTA-3’ and 5’-GGAGGATTCAAGGCAAGTTTTC-3’, respectively, used in the amplification polymerase chain reaction (PCR), were originally designed in the Primer3Plus program (https://primer3plus.com/cgi-bin/dev/primer3plus.cgi (accessed on 2 November 2020)). The potential formation of homo/heterodimers and hairpin structures were checked by the IDT OligoAnalyzer (https://eu.idtdna.com/calc/analyzer (accessed on 2 November 2020)); the specificity of selected primers was tested using NCBI Primer-BLAST (https://www.ncbi.nlm.nih.gov/tools/primer-blast/ (accessed on 2 November 2020)). One primer specific for single-base extension (SBE) reaction was designed for each single nucleotide polymorphism (SNP) (rs11003125, rs7096206, rs7095891, rs1800450, rs1800451, rs5030737) using a sequence from the NCBI dbSNP database (https://www.ncbi.nlm.nih.gov/projects/SNP/ (accessed on 2 November 2020)). Poly-T tails of different lengths were added to each SBE primer, and analysis of hairpin and dimer formation was performed. SBE primers were commercially synthesized and purified by high-performance liquid chromatography (Integrated DNA Technologies, Leuven, Belgium). The information about primers is listed in Table 2.

#### 2.3.2. Amplification PCR

Ten microliters of the PCR mixture contained a 1× *Taq* buffer with (NH_4_)_2_SO_4_ and without MgCl_2_, a 0.3 mM concentration of each deoxynucleotide (dNTP), a 0.25 μM concentration of each primer, 3 mM of MgCl_2_, 0.06 U of *Taq* DNA polymerase (Thermo Fisher Scientific, Waltham, MA, USA), and 2 μL of genomic DNA (of concentration 50 ng/μL). Cycling conditions were initial denaturation at 95 °C for 3 min., then 30 cycles of 95 °C for 30 s, 58 °C for 30 s and 72 °C for 90 s, followed by a final extension at 72 °C for 10 min. The formation of a correct PCR amplicon (1334 bp length) was checked by electrophoresis in 1% agarose gel after 1 hour at 90 V. Three microliters of the PCR mixture were treated with 5 U of Exonuclease I and 1.2 U of Shrimp Alkaline Phosphatase. The mixture was incubated at 37 °C for 1 hour, and the enzymes were inactivated by incubation at 85 °C for 15 min.

#### 2.3.3. Single-Base Extension (SBE) Reaction

Five microliters of the single-base extension (SBE) reaction mixture contained 0.5 μL of the SNaPshot Multiplex Ready Reaction Mix (ABI PRISM^®^ SNaPshot™ Multiplex kit, Applied Biosystems, Foster City, CA, USA), 2 μL of the PCR mixture after enzymatic treatment, and 1 μL of the SBE primer mix (concentrations of individual primers are listed in Table 2). Five microliters of positive and negative control reactions were prepared by mixing an appropriate volume of water, 0.5 μL of the SNaPshot Multiplex Ready Reaction Mix, and 0.5 μL of the SNaPshot Multiplex Control Primer Mix with and without 1 μL of the SNaPshot Multiplex Control Template, respectively. The SBE primer mix control reaction was prepared by mixing 3.5 μL of water and 0.5 μL of the SNaPshot Multiplex Ready Reaction Mix with 1 μL of the SBE primer mix without any template. Cycling conditions were 30 cycles at 96 °C for 10 s, 50 °C for 5 s, and 60 °C for 30 s. The SBE reaction mixture was treated with 1 U of Shrimp Alkaline Phosphatase and incubated at 37 °C for 1 h, which was followed by enzyme inactivation at 85 °C for 15 min.

#### 2.3.4. Capillary Electrophoresis

Zero point five microliters of the treated single-base extension (SBE) reaction mixture were mixed with 9.4 μL of Super-DI Formamide v2 and 0.1 μL of GeneScan™ 120 LIZ™ size standard (Applied Biosystems, Foster City, CA, USA). The mixture was denatured at 95 °C for 5 min, immediately chilled at −20 °C for 3 min, centrifuged and kept at 4 °C. The capillary electrophoresis was performed on an automated sequencer ABI 3130-Avant Genetic Analyzer (Applied Biosystems, Foster City, CA, USA) with a 22 cm long capillary (Applied Biosystems, Foster City, CA, USA) filled with NimaPOP-4 (NimaGen, Nijmegen, The Netherlands). The injection at 1.0 kV lasted 10 s. The electrophoresis was performed at 15 V, 60 °C, and 5 μA for 360 s. The laser was set at a constant power of 15 mW. Data were collected by the Data Collection software (version 3.0, Applied Biosystems, Foster City, CA, USA) and the results were analyzed using the GeneMapper ID software (version 3.2, Applied Biosystems, Foster City, CA, USA).

### 2.4. Sanger Sequencing

Eighty-three samples randomly chosen from the pool were sequenced using a Big Dye™ Terminator v3.1 Cycle Sequencing kit (Applied Biosystems, Foster City, CA, USA). PCR products treated with Exonuclease I and Shrimp Alkaline Phosphatase were used as templates, and an ethanol/EDTA precipitation procedure was used for purification of sequencing reactions. The capillary electrophoresis was performed on an automated sequencer ABI 3130-Avant Genetic Analyzer with a 36 cm long capillary filled with NimaPOP-7 (NimaGen, Nijmegen, The Netherlands). Data were collected by the Data Collection program and results were analyzed using the Sequencing Analysis program (version 5.4, Applied Biosystems, Foster City, CA, USA).

### 2.5. Calculation of Haplogenotype Combinations

An algorithm for calculating haplogenotype combinations (Haplogenotype Calculator) from determined genotypes was developed. Combinations from 2 to 10 single nucleotide polymorphisms (SNPs) were calculated. When individual boxes (Homozygote 1, Heterozygote, Homozygote 2) were filled with particular genotypes, the program created all possible combinations of these genotypes (background data set) and each combination in the background data set had a unique number. During the next step, the program compared the genotype combination of the real sample (determined experimentally) with the background data set. When a match was found, the unique number of the corresponding genotype combination was entered into the output datasheet. The whole sample set was analyzed this way. When 4 SNPs are investigated, for example, 11 possible combinations are calculated (such as SNP1+SNP2, SNP1+SNP3, SNP1+SNP4, SNP2+SNP3, SNP2+SNP4, SNP3+SNP4; SNP1+SNP2+SNP3, SNP1+SNP2+SNP4, SNP1+SNP3+SNP4, SNP2+SNP3+SNP4; SNP1+SNP2+SNP3+SNP4). When 10 SNPs are calculated, the application generates all allowed combinations of 2–10 SNPs. This algorithm has been made freely available as an online application on the URL http://78.128.251.116/ (accessed on 2 November 2020). The website is hosted by MetaCentrum VO (https://metavo.metacentrum.cz/ (accessed on 2 November 2020)). First, the spreadsheet containing the specific diagram must be downloaded from the main page of the website (link “scheme”). Then, an XML (Extensible Markup Language) file must be generated from this spreadsheet containing sample IDs and experimentally determined genotypes of SNPs. The XML file is uploaded to the application, the number of SNPs is selected, individual fields are filled with the allele combination for each SNP, and the calculation is run. The output datasheet is saved as an *.xlsx file. According to the number of analyzed SNPs, the output file contains separated sheets with combinations of SNP pairs, SNP triplets, SNP quadruplets etc. The calculation time depends on the number of SNPs and especially on the number of analyzed samples in the sample pool; it can take up to 1 hour. A step-by-step manual on the use of this application for calculating haplogenotypes is available on the main page of the website.

### 2.6. Statistical Analysis

Hardy-Weinberg equilibrium, linkage disequilibrium, minor allele frequencies and haplotype frequencies were calculated in the Haploview software (version 4.2, Broad Institute, Cambridge, MA, USA) [19].

## 3. Results

### 3.1. Design and Optimization of the SNaPshot Assay

The SNaPshot assay was developed and optimized for simultaneous detection of six single nucleotide polymorphisms (SNPs) in the *MBL2* gene. PCR was designed to amplify a single amplicon carrying all SNPs of interest. The PCR was optimized by adjusting the concentrations of the primers (0.3–0.5 μM) and MgCl_2_ (1–6 mM) and the cycling conditions (gradient of T_a_ 54–62 °C, with a 3–5 min. duration of initial denaturation, for 25–45 cycles). To minimize reaction costs, the volume of the SNaPshot Multiplex Ready Reaction Mix added to the single-base extension (SBE) reaction was reduced, as well as the total volumes of the SBE and PCR reactions. Considering the positions of three polymorphisms in the exon 1, which are very close to each other, two out of six specific primers used in the SBE reaction were synthesized with degenerate bases incorporated into their sequence (Table 2).

Six singleplex SBE reactions were performed with a PCR template to determine the position of the peaks of individual SNP alleles in the electropherogram and to check for peaks caused by non-specific annealing. A multiplex SBE reaction with the SBE primer mix, and without the PCR template, served to check for the formation of heterodimer products. Overlapping of the peaks of the three shortest SBE primers was observed after analysis of the multiplex SBE reaction; nevertheless, it was still possible to determine genotypes due to the color difference of incorporated bases belonging to the particular SNPs (Figure 1). Concentrations of specific primers were adjusted, and the number of SBE reaction cycles was increased to obtain more balanced peak heights.

Finally, the panel and bins for allele identification were entered into the GeneMapper program to enable the automatic analysis.

### 3.2. Mannose-Binding Lectin Gene (MBL2) Genotyping

Altogether, 328 samples were successfully genotyped by the SNaPshot assay. Frequencies of the variant alleles of all six *MBL2* gene single nucleotide polymorphisms (SNPs) (rs1800451, rs1800450, rs5030737, rs7095891, rs7096206, rs11003125) were in accordance with the Hardy-Weinberg equilibrium (*p* = 1.00, 0.73, 0.80, 0.12, 0.38, 0.74, respectively). The pairwise linkage disequilibrium (LD) was calculated, and all studied *MBL2* SNPs were in the LD block (Appendix A). Minor allele frequencies (MAFs) in the Czech population corresponded to MAFs from the 1000 Genomes Project Phase 3 (NCBI database dbSNP) for the European population. MAFs of all studied *MBL2* SNPs in other populations also have been included in Table 3 for comparison. Calculated haplotype frequencies were similar to those reported in the Danish population (Table 4).

### 3.3. Validation of the SNaPshot Assay Results

To validate results obtained by the developed SNaPshot assay, we used classical Sanger sequencing of 83 randomly selected samples. A perfect match (100% fit) was found between genotypes determined by Sanger sequencing and the SNaPshot assay.

### 3.4. Calculation of Haplogenotype Combinations

The Haplogenotype Calculator created within the scope of this study was used for calculation of haplogenotype combinations from six single nucleotide polymorphisms (SNPs) in the *MBL2* gene determined in 328 samples. Distribution of the *MBL2* haplogenotypes calculated from all six SNPs among samples from healthy individuals is listed in Table 5.

## 4. Discussion

Here, the SNaPshot assay was introduced for simultaneous genotyping of the six functional polymorphisms within the *MBL2* gene. Similar assays have recently been used but none of these was designed for genotyping all six single nucleotide polymorphisms (SNPs) [21,22,23,24]. Due to the proximity of the individual analyzed SNPs, we performed a singleplex PCR reaction, which subsequently served as a template for the single-base extension (SBE) reaction. We have achieved consistent results within the optimized SBE reaction, even with a significantly minimized volume (0.5 μL) of the commercially used mix. Regarding a small portion of analyzed samples (4.9%), non-specific peaks in electropherograms were observed. These peaks were at the same positions and colors as peaks of alleles D and Q, which could be potentially problematic during the consequent analysis. However, we eliminated the risk of inaccurate allele identification by setting the heterozygote balance (the min peak height ratio) in the GeneMapper software. All the problematic samples were stored at +4 °C after DNA isolation for more than six months, or their stock solutions were stored at −20 °C for more than two years. Thus, it is very likely that these samples contained partially degraded DNA, which could adversely affect the SNaPshot assay effectiveness with potential amplification bias.

The choice of an appropriate method for genotyping SNPs in clinical or research laboratories depends on several factors, including the number and type of samples and the available quantity/quality of the analyzed DNA. Also, financial, material, and instrumentation requirements, or the availability of a commercial genotyping service, are considered. During *MBL2* genotyping, the proximity of the polymorphisms in the *MBL2* exon 1 is a crucial factor that must be taken into account in selecting the method. A combination of methods suitable for determining SNPs in the promoter and the first exon often needs to be applied in *MBL2* genotyping [25,26].

Up to this day, a wide spectrum of other methods have been used for detection of *MBL2* gene variants, including commercially available TaqMan assays [27], high-resolution melt analysis (HRMA) [28], reverse hybridization with membrane-immobilized sequence-specific oligonucleotide probes (reverse PCR-SSOP) [29], PCR with sequence-specific primers (SSP-PCR) [30,31], PCR and restriction-fragment length polymorphism (PCR-RFLP) analysis [32], an oligonucleotide ligation assay [33,34], a single-strand conformation polymorphism technique (PCR-SSCP) [35], the iPLEX assay on a MassArray system [36], Sanger sequencing [37], and pyrosequencing [38].

Genotyping methods based on PCR or enzymatic digestion of specific PCR products with analysis of genotypes on agarose/acrylamide gel are widely used due to their simplicity and low requirement for materials and instrumentation. Conversely, these techniques are laborious and, in effect, not suitable for high-throughput genotyping. Furthermore, the low sensitivity and robustness of these methods might pose a problem. A TaqMan assay, which belongs to high-throughput methods, is quite simple, fast, and sensitive. However, analyses of interrogated polymorphisms are performed in separate reactions, leading to a higher consumption of the template DNA. The assay can be multiplexed but, in view of the proximity of the individual SNPs in exon 1 of *MBL2* it can be very difficult, if not impossible. This could lead to the need for several individual reactions, which would render the analysis expensive.

Conversely, the proximity of SNPs in exon 1 become an advantage when the pyrosequencing method is used for the genotyping of these SNPs. Pyrosequencing and HRMA are very rapid and sensitive methods; however, four (pyrosequencing) and five (HRMA) analyses, respectively, are needed to determine six *MBL2* polymorphisms, which increases the consumption of the template DNA.

Unlike these methods, the SNaPshot minisequencing method is at present the only method capable of reliable automated determination of the complete *MBL2* haplogenotype in a single analysis and is less prone to heterozygosity loss than Sanger sequencing [39]. The high sensitivity and short analysis time also count among the benefits of this method [40]. The comparison of methods used for *MBL2* genotyping is summarized in Table 6.

During our study, 328 samples from healthy individuals from the Czech population were genotyped using the SNaPshot assay. Minor allele frequencies (MAFs) of all six *MBL2* SNPs in the healthy Czech population were similar to the MAFs deposited in the database for the European population (Table 3). The calculated haplotype frequencies were consistent with those determined in the Danish [20] and Czech [42] populations (Table 4). A rare secretor haplotype LYQB was detected in the sample pool. To our knowledge, it is the first evidence of this haplotype in the Czech population. Additionally, *MBL2* haplogenotypes from six SNPs were determined in the Czech population for the first time (Table 5).

Concerning some cases, diseases can be associated with haplotypes (i.e., with the sequential allele arrangement of particular SNPs) or diplotypes (pairs of haplotypes) rather than alleles or genotypes alone [47]. Furthermore, the frequency of alleles can be in a strong Hardy-Weinberg Disequilibrium in sample sets of subjects with investigated diseases. Regarding such instances, diplotype-based association analyses are more powerful than haplotype-based analyses [48]. Methods used for genotyping are unable to determine haplotypes, although methods capable of identifying diplotypes have been developed [49]. Therefore, several programs for haplotype phasing (haplotype estimation from determined genotypes) [50,51], calculation of haplotype frequencies and association tests [52,53], and diplotype-based association analysis [54] have been developed.

Conversely, haplogenotypes (combinations of unphased genotypes) have been used in association studies [55]. Here, a simple online application for the calculation of haplogenotypes by combining determined genotypes of two to ten SNPs has been prepared to facilitate further statistical analyses.

## 5. Conclusions

A novel SNaPshot assay for detection of six single nucleotide polymorphisms (SNPs) in the *MBL2* gene was developed, optimized, and used for the genotyping of 328 samples from healthy individuals from the Czech population. Results of the assay were validated through comparison with results of sequencing analysis, which were concordant. Calculated minor allele frequencies (MAFs) were in accordance with MAFs for the European population deposited in the database. This high-throughput, cost-effective, and accurate method, therefore, can be used in further association studies as well as in clinical practice. The main limitation of the method is the quality of the DNA in tested samples. Moreover, the freely available online application developed for this study can be employed for the calculation of haplogenotype combinations from up to 10 genotypes.

## Figures and Tables

**Figure 1 diagnostics-11-00301-f001:**
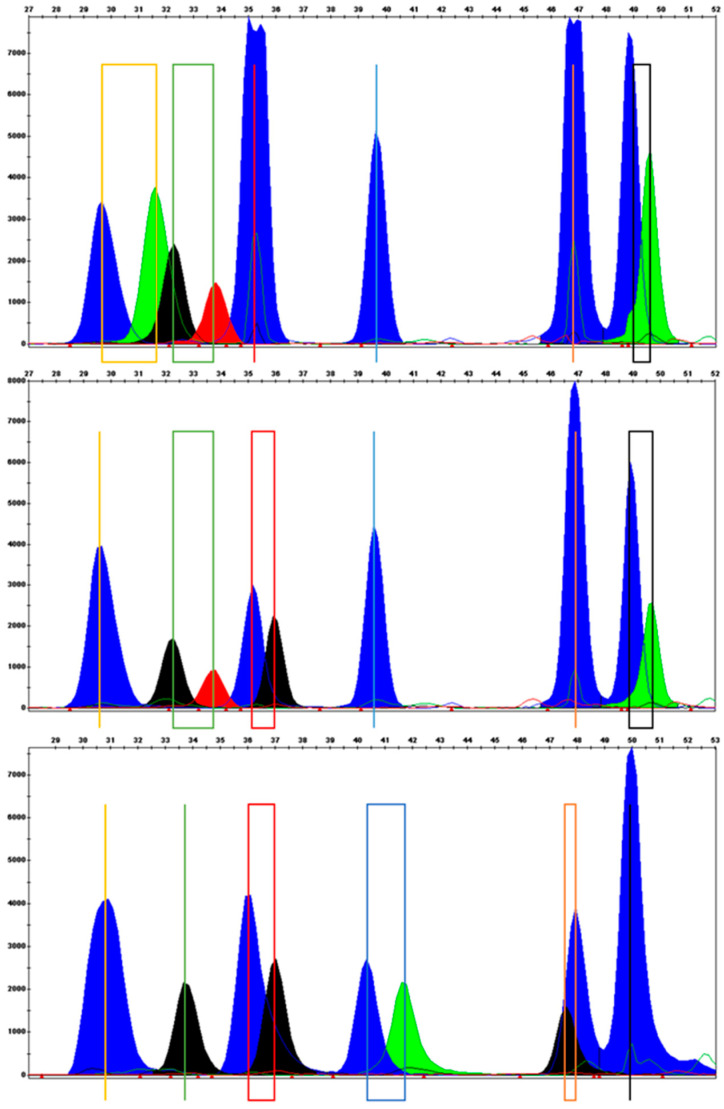
Electropherograms of three samples showing peaks of all alleles. Alleles of individual single nucleotide polymorphisms (SNPs) are highlighted by color line or rectangle (yellow–allele A/B, green–allele A/C, red–allele X/Y, blue–allele A/D, orange–allele H/L, black–allele P/Q).

**Table 1 diagnostics-11-00301-t001:** Demographic data of 328 individuals with good systemic health who enrolled in this study; all were selected from the Czech population.

Characteristics	*n* = 328 Total	*n* = 192 Female	*n* = 136 Male
Age (mean ± SD, median, IQR) in years	34.0 ± 13.5, 30.0, 23.0–43.0	34.0 ± 14.3, 27.0, 22.0–44.5	34.1 ± 12.2, 31.0, 25.0–41.0

IQR, interquartile range; *n*, number of subjects; SD, standard deviation.

**Table 2 diagnostics-11-00301-t002:** Primers used in the single-base extension (SBE) reaction for determining the mannose-binding lectin gene (*MBL2*) single nucleotide polymorphisms (SNPs).

Rs Number	Nucleotide Substitution ^1^ (Location, Amino Acid Substitution/SNP Position ^2^)	Allele	Primer Sequence (5’→ 3’)	DNA Strand	Concentration of the Primer in the SBE Primer Mix (μM)
rs1800450	161G>A (exon 1, Gly54Asp)	A/B	CCAGGCAAAGATGGGYGTGATG ^3^	forward	0.1
rs1800451	170G>A (exon 1, Gly57Glu)	A/C	TTTTACGTACCTGGTTCCCCCTTTTCT	reverse	0.18
rs7096206	−290C>G (promoter, -221 ^2^)	X/Y	TTTTTTTTGGTCCCATTTGTTCTCACTGCCAC	forward	0.06
rs5030737	154C>T (exon 1, Arg52Cys)	A/D	TTTTTTTTTTTTCCCTTTTCTYCCTTGGTGYCATCAC ^3^	reverse	0.08
rs11003125	−619C>G (promoter, -550 ^2^)	H/L	TTTTTTTTTTTTTTTTGGAGTTTGCTTCCCCTTGGTGTTTTA	reverse	0.08
rs7095891	−66C>T (5’ UTR of exon 1, +4 ^2^)	P/Q	TTTTTTTTCAGGGAAGGTTAATCTCAGTTAATGAACACATATTTACC	reverse	0.06

^1^ Single nucleotide polymorphism (SNP) position according to the database NCBI dbSNP HGVS (Human Genome Variation Society) names NM_000242.2:c entry; ^2^ SNP position used in the literature, e.g., Garred et al. [8] or Świerzko and Cedzyński [18]; ^3^ Letter Y in the primer sequence indicates a site containing a degenerate base (cytosine or thymine). Single-base extension (SBE).

**Table 3 diagnostics-11-00301-t003:** Minor allele frequencies (MAFs) of mannose-binding lectin gene (*MBL2*) single nucleotide polymorphisms (SNPs) in a genotyped sample pool of Czech healthy individuals and in European (EUR), African (AFR), East Asian (EAS), South Asian (SAS) and American (AMR) populations according to NCBI database dbSNP (https://www.ncbi.nlm.nih.gov/snp/ (accessed on 2 November 2020)).

rs Number	Allele	MAF (%)					
		*n* = 328 Czech	*n* = 1006 EUR	*n* = 1322 AFR	*n* = 1008 EAS	*n* = 978 SAS	*n* = 694 AMR
rs11003125	H	38.3	38.6	8.7	44.6	31.0	40.0
rs7096206	X	21.4	22.1	15.4	18.6	28.0	13.0
rs7095891	Q	23.6	19.8	54.0	13.8	25.0	17.0
rs5030737	D	8.5	6.0	0.2	0.1	5.0	3.0
rs1800450	B	13.5	14.1	1.4	14.8	15.0	22.0
rs1800451	C	1.8	1.2	25.9	0.0	4.0	2.0

*n*, number of subjects.

**Table 4 diagnostics-11-00301-t004:** Mannose-binding lectin gene (*MBL2*) haplotype frequencies in a genotyped sample pool of 328 Czech and 100 Danish healthy individuals.

*MBL2* Haplotype ^1^	Combination of Alleles (Secretor Haplotype)	MBL Phenotype ^2^	Frequency in Czech Population (%)	Frequency in Danish Population ^3^ (%)
G G C C G G	HYPA	High	29.8	28.5
C G T C G G	LYQA	High	21.7	23.5
C C C C G G	LXPA	Low	21.4	19.5
C G C C A G	LYPB	Undetectable	13.3	13.5
G G C T G G	HYPD	Undetectable	8.5	8.5
C G C C G G	LYPA	Intermediate	3.3	4.5
C G T C G A	LYQC	Undetectable	1.8	2.0
C G T C A G	LYQB	Undetectable	0.2	-

^1^ rs11003125, rs7096206, rs7095891, rs5030737, rs1800450, rs1800451; ^2^ According to Cedzyński et al. and Verdu et al. [4,11]; ^3^ According to Steffensen et al. [20].

**Table 5 diagnostics-11-00301-t005:** Mannose-binding lectin gene (*MBL2*) haplogenotype distribution among samples from 328 healthy Czech individuals.

*MBL2* Haplogenotype ^1^	Combination of Secretor Haplotypes ^2^	Frequency (%)
GC GG CT CC GG GG	HYPA/LYQA	16.2
GC GC CC CC GG GG	HYPA/LXPA	11.9
CC GC CT CC GG GG	LYQA/LXPA	10.1
GG GG CC CC GG GG	HYPA/HYPA	7.9
GC GG CC CC GA GG	HYPA/LYPB	7.3
CC GG CT CC GA GG	LYQA/LYPB	6.1
CC CC CC CC GG GG	LXPA/LXPA	5.5
GG GG CC CT GG GG	HYPA/HYPD	5.2
CC GC CC CC GA GG	LXPA/LYPB	4.0
CC GG TT CC GG GG	LYQA/LYQA	3.7
GC GG CC CT GA GG	LYPB/HYPD	3.0
GC GC CC CT GG GG	LXPA/HYPD	3.0
GC GG CT CT GG GG	LYQA/HYPD	2.4
GC GG CC CC GG GG	HYPA/LYPA	2.4
CC GG CC CC AA GG	LYPB/LYPB	2.1
CC GC CT CC GG GA	LXPA/LYQC	1.8
CC GG CC CC GA GG	LYPB/LYPA	1.2
CC GG CT CC GG GG	LYQA/LYPA	1.2
GC GG CC CT GG GG	HYPD/LYPA	0.9
CC GC CC CC GG GG	LXPA/LYPA	0.9
GG GG CC TT GG GG	HYPD/HYPD	0.9
GC GG CT CC GG GA	HYPA/LYQC	0.6
CC GG CT CC GA GA	LYPB/LYQC	0.6
CC GG TT CC GG GA	LYQA/LYQC	0.3
GC GG CT CT GG GA	HYPD/LYQC	0.3
GC GG CT CC GA GG	HYPA/LYQB	0.3

^1^ rs11003125, rs7096206, rs7095891, rs5030737, rs1800450, rs1800451; ^2^ Translated from the haplogenotypes.

**Table 6 diagnostics-11-00301-t006:** Comparison of methods used for mannose-binding lectin gene (*MBL2*) genotyping.

	Allele -Specific PCR (AS-PCR)	ARMS ^3^/Double ARMS ^3^ (+ Multiplex Allele -Specific PCR)	PCR and Restriction-Fragment Length Polymorphism (PCR-RFLP)	Commercial TaqMan Assay ^5^	High-Resolution Melt Analysis (HRMA)	Commercial INNO-LiPA MBL2 kit (Reverse PCR-SSOP)	Pyro- Sequencing	Sanger Sequencing	*MBL2* SNaPshot Assay
**principle of allele discrimination/detection**	PCR with a primer specific for one allele	PCR with primers specific for both alleles	allele-specific enzymatic cleavage of PCR amplicon	allele-specific hybridization of fluorescently labelled probe	temperature-dependent allele-specific hybridization of fluorescently labelled probe	hybridization of biotinylated PCR product with membrane immobilized sequence-specific oligonucleotide probes	chemiluminiscence-based detection of nucleotides during sequencing-by-synthesis reaction	detection of the sequence of an oligonucleotide amplified in PCR with fluorescently labelled dideoxyribonucleotides	allele-specific SBE by a single fluorescently labelled dideoxyribonucleotide (minisequencing)
**post-PCR analysis**	yes	yes	yes	no	no, when real-time PCR thermocycler is used	yes	no	yes	yes
**analysis time**	2 h ^2^	2–3 h ^2^	2 h + 1–3 h ^4^	1–2 h ^6^	1–1.5 h + 2–8 min. ^8^	3–4 h	2–3 h	6–7 h	5–6 h
**number of work steps**	2 (PCR, gel analysis)	2 (PCR, gel analysis)	4 (PCR, gel analysis, RFLP, gel analysis)	1 (real-time PCR)	1 (when real-time PCR thermocycler is used for PCR and subsequent melting temperature analysis)	9 (PCR, gel analysis, denaturation, hybridization, 2 washing steps, 3-step color development)	4 (PCR, gel analysis, purification, pyrosequencing)	5 (PCR, enzymatic cleaning, sequencing reaction, purification, analysis on sequencer)	5 (PCR, enzymatic cleaning, SBE reaction, enzymatic cleaning, analysis on sequencer)
**automatic analysis**	no	no	no	yes	yes	no	yes	yes	yes
**number of analyses for complete *MBL2* haplogenotype ^1^**	12	6	6	6	5	1	4 ^9^	2	1
**number of oligonucleotide primers + labelled primers/probes** **for complete *MBL2* haplogenotype ^1^**	24 primers	15 primers	6 primers	6 TaqMan assays (12 primers + 12 TaqMan probes)	10 primers + 5 TaqMan probes	4 primers	8 primers + 4 biotinylated primers	2 primers	8 primers
**estimated cost of analysis of whole haplogenotype ^1^**	1 USD	1 USD	2 USD	2 USD	1 USD	product was discontinued	2 USD	5 USD	1.50 USD
**input amount** **of template DNA**	20–200 ng	20–200 ng	50–500 ng	1–20 ng	10–20 ng	200–500 ng	10–100 ng	10–250 ng	10–100 ng
**assay robustness**	low	low	low–medium	medium–high	high	low–medium	medium	medium–high	medium–high
**special equipment requirement**	-	-	-	real-time PCR thermocycler	real-time PCR thermocycler or fluorescence scanning/detection system	water bath with shaking platform, aspiration apparatus	vacuum prep workstation, pyrosequencing machine	automated DNA sequencer	automated DNA sequencer
**SNP genotyping throughput**	low	low	low	high	high	medium	high	high	high
**software for automatic allele calling**	no	no	no	yes (SDS software, SNPman program) ^7^	yes real-time PCR instruments with HRMA compatible software with genotype auto-calling function	no	no	yes (Mutation Surveyor, GeneMarker, Minor Variant Finder Software, SeqScape™ Software, Variant Reporter™ Software) ^10^	yes (GeneMapper, GeneMarker) ^11^
**Ref.**	[41]	[30,42]	[43,44]	[27]	[28]	[29]	[45]	[37]	-

^1^ rs11003125, rs7096206, rs7095891, rs5030737, rs1800450 and rs1800451; ^2^ Analysis time depends on polymerase chain reaction (PCR) length and gel concentration; ^3^ ARMS–amplification refractory mutation system; ^4^ Separation time depends on gel concentration and the length of cleaved fragments; ^5^ TaqMan^®^ SNP Genotyping Assays (Applied Biosystems), probes: C__11876879_10 (rs11003125), C__27858274_10 (rs7096206), C__26813436_10 (rs7095891), C___2336610_10 (rs5030737), C___2336609_20 (rs1800450) and C___2336608_20 (rs1800451); ^6^ depends on number of cycles; ^7^ Sequence Detection Software (SDS) by Applied Biosystems™ (www.thermofisher.com (accessed on 2 November 2020)), SNPman program by Konopac et al. [46]; ^8^ Time of temperature melt analysis depends on a temperature range and thermal ramp rate of the instrument; ^9^ Due to the proximity of the three single nucleotide polymorphisms (SNPs) in exon 1 only one analysis is needed to determine these SNPs; ^10^ Mutation Surveyor^®^ and GeneMarker^®^ by SoftGenetics^®^ (https://softgenetics.com (accessed on 2 November 2020)). Minor Variant Finder Software, SeqScape™ Software and Variant Reporter™ Software by Applied Biosystems™ (www.thermofisher.com (accessed on 2 November 2020)); ^11^ GeneMapper™ by Applied Biosystems™ (www.thermofisher.com (accessed on 2 November 2020)), GeneMarker^®^ by SoftGenetics^®^ (https://softgenetics.com (accessed on 2 November 2020)). Single-base extension (SBE). Mannose-binding lectin gene (*MBL2*). Reverse hybridization with membrane-immobilized sequence-specific oligonucleotide probes (reverse PCR-SSOP).

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
