# Peer review of "A SNaPshot Assay for Determination of the Mannose-Binding Lectin Gene Variants and an Algorithm for Calculation of Haplogenotype Combinations"

_diagnostics, 2021, doi:10.3390/diagnostics11020301_

Round 1
Reviewer 1 Report
The authors describe a technique and an algorithm useful to calculate combinations of MBL2 gene haplotypes, whose variants are associated with infections and autoimmune diseases. The authors define their work as high-throughput, however many aspects need to be defined and better described.
MAJOR REVISION:
1) The introduction is poor, not well explanatory and with lines 46-52 redundant.
2) The first part of the discussion section (lines 195-206) must be moved to the introduction.
3) The conclusion of the introduction is confusing
4) "study subject and sampling" section: a detailed table with the demographic and clinical characteristics of the individuals enrolled in the study is missing.
5) Table 4 is mentioned as before in the text. Tables 1, 2 and 3 are instead missing within the text.
6) At the conclusion of the introduction the authors mention 328 individuals analyzed, however this number is not clear in the materials and methods. Moreover, why the authors analyze only 83 patients with Sanger sequencing? The validation must be more accurate.
7) A descriptive and punctual part of the results obtained is missing.
8) In the discussion section:
line 200: “and variation of their frequencies among ethnic groups” specify the meaning of this sentence in this context.
line 218, the authors declare that the assay has been optimized: please describe accurately all the steps that have been optimized, as part of results.
Line 235 some sample ...how many??? This is not an acceptable scientific language
Line 263 “Results of the assay were validated through comparison with results of sequencing analysis, which were concordant.” The presented data do not confirm this judgment
9) In light of the avant-garde techniques available today to perform DNA analysis, the limitation of the SNaPshot assay in giving reliable results in case of degraded DNA is an objective problem. The authors should technically improve this aspect since they propose their method as a useful tool both for research and diagnostics.
10) The number of individuals studied is too small in relation to the final goal of the authors and limited to a single population.
MINOR REVISION
Please revise the nomenclature according to HGVS. i.e: Alleles B (rs1800450), contain substitution of Gly54Asp (+230G>A). HGVS NM_001378373.1:c.161G>A, NP_000233.1:p.Gly54Asp, C (rs1800451) Gly57Glu (+239G>A): NM_001378373.1:c.170G>A, NP_000233.1:p.Gly57Glu; D (rs5030737) Arg52Cys 56(+223C>T): HGVS NM_001378373.1:c.154C>T, NP_000233.1:p.Arg52Cys
Author Response
Referee 1
MAJOR REVISION:
1) The introduction is poor, not well explanatory and with lines 46-52 redundant.
Answer:
The Introduction section was modified and re-written to be more explanatory (lines 41-74). Redundant information was deleted.
Mannose-binding lectin (MBL, also known as mannose-binding protein, mannan-binding protein/lectin, Collectin 1, MBP1, or mannose-binding protein C) is a crucial part of the innate immunity. It is classified as an acute-phase protein because its plasma level increases after the onset of inflammation. MBL belongs to a group of soluble pattern-recognition receptors that can recognize pathogen-associated molecular patterns and damage-associated molecular patterns released from activated cells, cells undergoing apoptosis, necrosis, or host cells with altered surface glycosylation patterns related to cancer phenotypes [1–5]. MBL trimers and tetramers are the forms most common in blood circulation due to their highest effectiveness in complement activation after the interaction with MBL-associated serine proteases (MASP 1, -2, -3) [6–9]. Besides, MBL works as an opsonin and enables phagocytosis independently on the complement [1,4].
The concentration of the functional MBL in blood circulation is affected by the MBL2 gene variant located on chromosome 10q11.2-21. Alleles B (rs1800450, NM_000242.2:c.161G>A), C (rs1800451, NM_000242.2:c.170G>A) or D (rs5030737, NM_000242.2:c.154C>T) possessing single nucleotide polymorphisms (SNPs) in exon 1 cause amino acid substitution of Gly54Asp, Gly57Glu or Arg52Cys, respectively. These variants lead to the production of structurally impaired protein. As a result of the substitutions, higher MBL oligomers disintegrate more rapidly to lower oligomers, which are no longer capable of activating the complement system. These alleles are summarily called alleles O, whereas the allele without substitution (normal allele) is called allelle A. Alleles H/L (rs11003125) and Y/X (rs7096206) carrying SNPs localized in the promoter region (NM_000242.2:c.-619C>G and NM_000242.2:c.-290C>G, respectively), and allele P/Q (rs7095891) with the SNP in the 5' untranslated region (5'UTR) of the exon 1 (NM_000242.2:c.-66C>T), alter MBL2 transcription and translation, directly affecting MBL plasma level [8,10].
Because of strong linkage disequilibrium (LD) among the six SNPs and variation of frequencies of these SNPs among ethnic groups, only seven MBL secretor haplotypes (allele combinations) are relatively common. HYPA and LYQA haplotypes are associated with high MBL serum levels, LYPA with intermediate levels, and LXPA with low MBL production. Defect haplotypes LYPB, LYQC and HYPD are responsible for expressing undetectable amounts of MBL [10,11]. Nevertheless, the MBL level in circulation depends on a combination of two haplotypes; for example, the HYPA/LXPA haplogenotype is responsible for medium MBL level while LYPB, LYQC or HYPD combined with LYPA leads to low MBL levels in the serum [12].
MBL deficiency (a common human immunodeficiency, 5-30 % of the population is affected) has been associated with increased susceptibility towards infectious diseases and autoimmune diseases [13]. Determining genotypes of all six SNPs and resulting haplogenotypes is necessary for predicting the MBL phenotype.
2) The first part of the discussion section (lines 195-206) must be moved to the introduction.
Answer:
According to referee’s request the mentioned part of the text in the Discussion section was moved and incorporated to the text of the Introduction section in the revised manuscript (lines 63-75):
Because of strong linkage disequilibrium (LD) among the six SNPs and variation of frequencies of these SNPs among ethnic groups, only seven MBL secretor haplotypes (allele combinations) are relatively common. HYPA and LYQA haplotypes are associated with high MBL serum levels, LYPA with intermediate levels, and LXPA with low MBL production. Defect haplotypes LYPB, LYQC and HYPD are responsible for expressing undetectable amounts of MBL [10,11]. Nevertheless, the MBL level in circulation depends on a combination of two haplotypes; for example, the HYPA/LXPA haplogenotype is responsible for medium MBL level while LYPB, LYQC or HYPD combined with LYPA leads to low MBL levels in the serum [12].
MBL deficiency (a common human immunodeficiency, 5-30 % of the population is affected) has been associated with increased susceptibility towards infectious diseases and autoimmune diseases [13]. Determining genotypes of all six SNPs and resulting haplogenotypes is necessary for predicting the MBL phenotype.
3) The conclusion of the introduction is confusing.
Answer:
We found out that the manuscript originally submitted for review did not contain the Results section. This mistake was very likely caused by uploading of the wrong version of the manuscript. We apologize for this mistake. The lack of the Results section in the manuscript could have resulted in the conclusion of the introduction being confusing to the reader. The missing Results section was added to the revised manuscript (lines 234-297) and we hope that in combination with that section, the last paragraph of the Conclusion makes sense now.
4) "study subject and sampling" section: a detailed table with the demographic and clinical characteristics of the individuals enrolled in the study is missing.
Answer:
Only individuals who were generally healthy were enrolled in this study; therefore, the clinical characteristics of the individuals were in the physiological range for respective age and sex (no detailed laboratory examination was done). Inclusion criteria are described in the chapter “Study subjects and sampling” (lines 98-102). Information about smoking status and demographic characteristics were added to the text of the chapter (lines 103-106):
328 individuals with good systemic health (mean age ± standard deviation of 34.0 ± 13.5 years, 192 women and 136 men) were enrolled in this study; for demographic data, see Table 1. 1.
A table containing demographic data was added to the chapter “Study subjects and sampling” in the section Materials and methods (Table 1):
Table 1. Demographic data of 328 individuals with good systemic health who were enrolled in this study; all were selected from the Czech population.
|
Characteristics |
N=328 Total |
N=192 Female |
N=136 Male |
|
Age (mean ± SD, median, IQR) in years |
34.0 ± 13.5, 30.0, 23.0-43.0 |
34.0 ± 14.3, 27.0, 22.0-44.5 |
34.1 ± 12.2, 31.0, 25.0-41.0 |
IQR, interquartile range; N, number of subjects; SD, standard deviation.
5) Table 4 is mentioned as before in the text. Tables 1, 2 and 3 are instead missing within the text.
Answer:
We apologize for these mistakes. The mistakes were corrected (line 141, 166, 333, 334, and 337).
6) At the conclusion of the introduction the authors mention 328 individuals analyzed, however this number is not clear in the materials and methods. Moreover, why the authors analyze only 83 patients with Sanger sequencing? The validation must be more accurate.
Answer:
The information about total number of the analyzed individuals was added to the chapter “Study subjects and sampling” within the section Materials and methods (lines 105-106):
328 individuals with good systemic health (mean age ± standard deviation of 34.0 ± 13.5 years, 192 women and 136 men) were enrolled in this study; for demographic data, see Table 1. 1.
The Sanger sequencing was used for validation of the SNaPshot assay accuracy. 83 samples randomly chosen from the sample pool genotyped by the SNaPshot assay were analyzed by the Sanger sequencing and no difference between sequencing results and result obtained by the SNaPshot assay was observed. Based on this result, it was concluded that the SNaPshot method was not associated with any systematic error. Moreover, the pool of sequenced samples contained all allele combinations of the SNPs. Therefore, it was decided that the number of samples sequenced for validation is sufficient and sequencing of additional samples would be cost- and time-consuming, without adding any further value to the validation.
7) A descriptive and punctual part of the results obtained is missing.
Answer:
We agree with the referee. The missing Results section was added to the revised manuscript (lines 234-297) and we apologize for this mistake.
8) In the discussion section:
- Line 200: “and variation of their frequencies among ethnic groups” specify the meaning of this sentence in this context.
Answer:
We thank the referee for this remark. The sentence was re-formulated in the text which was moved to the Introduction section (lines 63-65):
Because of strong linkage disequilibrium (LD) among the six SNPs and variation of frequencies of these SNPs among ethnic groups, only seven MBL secretor haplotypes (allele combinations) are relatively common.
- Line 218, the authors declare that the assay has been optimized: please describe accurately all the steps that have been optimized, as part of results.
Answer:
The missing Results section with chapter “Design and optimization of the SNaPshot assay” describing the optimization process (lines 235-256) was added to the revised manuscript. We apologize for this mistake.
- Line 235 some sample ...how many??? This is not an acceptable scientific language
Answer:
The paragraph containing this sample description was re-written to clarify the information about the problematic samples (lines 317-325):
4.9% of samples analyzed in this study showed non-specific peaks in their electropherograms. Non-specific peaks appeared in the same positions and colors with peaks belonging to alleles D and Q, which might be problematic during the analysis. However, we eliminated the risk of inaccurate allele calling by setting heterozygote balance (the min peak height ratio) in the GeneMapper software. All the problematic samples were stored at +4°C for more than six months after DNA isolation or their stock solutions were stored at -20°C for more than two years. It is therefore very likely that these samples have already undergone partial DNA degradation.
- Line 263 “Results of the assay were validated through comparison with results of sequencing analysis, which were concordant.” The presented data do not confirm this judgment
Answer:
As the first submitted manuscript lacked the Results section, the presented data were not sufficient. The missing Results section containing the chapter concerning the assay validation (lines 283-286) was added to the revised manuscript.
9) In light of the avant-garde techniques available today to perform DNA analysis, the limitation of the SNaPshot assay in giving reliable results in case of degraded DNA is an objective problem. The authors should technically improve this aspect since they propose their method as a useful tool both for research and diagnostics.
Answer:
The MBL2 SNaPshot assay is not suitable for analysis of degraded DNA as it was not designed for this type of samples. The authors’ intention was to emphasize the fact that the quality of the DNA sample (partially degraded DNA) can be a potential limitation of this method. The particular paragraph in the Discussion section was re-formulated to clarify the information (lines 317-329):
4.9% of samples analyzed in this study showed non-specific peaks in their electropher-ograms. Non-specific peaks appeared in the same positions and colors with peaks belonging to alleles D and Q, which might be problematic during the analysis. However, we eliminated the risk of inaccurate allele calling by setting heterozygote balance (the min peak height ratio) in the GeneMapper software. All the problematic samples were stored at +4°C after DNA isolation for more than six months or their stock solutions were stored at -20°C for more than two years. It is therefore very likely that these samples have already undergone partial DNA degradation. The amplicon's length could bring potential problems in the analysis of partially degraded DNA samples. Similarly, partially degraded DNA (e.g. long-stored samples) could adversely affect the SNaPshot assay effectiveness with potential amplification bias. Therefore, this non-specific amplification resulting in the non-specific peaks can be eliminated by reducing the total number of cycles in the SBE reaction.
10) The number of individuals studied is too small in relation to the final goal of the authors and limited to a single population.
Answer:
The methodology of genotyping is not population-dependent. Minor allele frequency (MAF) of particular SNP corresponds to MAF in European population even if the occurrence of the SNP in the population is less than 10%. The authors are of the opinion that 328 healthy individuals from the Czech population is a sufficient number to compare the results of genotyping with results for European or the Danish population.
MINOR REVISION:
Please revise the nomenclature according to HGVS. i.e: Alleles B (rs1800450), contain substitution of Gly54Asp (+230G>A). HGVS NM_001378373.1:c.161G>A, NP_000233.1:p.Gly54Asp, C (rs1800451) Gly57Glu (+239G>A): NM_001378373.1:c.170G>A, NP_000233.1:p.Gly57Glu; D (rs5030737) Arg52Cys 56(+223C>T): HGVS NM_001378373.1:c.154C>T, NP_000233.1:p.Arg52Cys
Answer:
The nomenclature of the SNPs was modified according to the NM_000242.2:c entry in the database NCBI dbSNP HGVS names (lines 51-61):
Alleles B (rs1800450, NM_000242.2:c.161G>A), C (rs1800451, NM_000242.2:c.170G>A) or D (rs5030737, NM_000242.2:c.154C>T) possessing single nucleotide polymorphisms (SNPs) in exon 1 cause amino acid substitution of Gly54Asp, Gly57Glu or Arg52Cys, respectively. These variants lead to the production of structurally impaired proteins. As a result of the substitutions, higher MBL oligomers disintegrate more rapidly to lower oligomers, which are no longer capable of activating the complement system. These alleles are together called O, whereas the allele without substitution (normal allele) is called A. Alleles H/L (rs11003125) and Y/X (rs7096206) carrying SNPs localized in the promoter region (NM_000242.2:c.-619C>G and NM_000242.2:c.-290C>G, respectively), and allele P/Q (rs7095891) with the SNP in the 5' untranslated region (5'UTR) of the exon 1 (NM_000242.2:c.-66C>T), alter MBL2 transcription and translation, directly affecting MBL plasma level [8,10].
The nomenclature was also corrected in the table containing information on SBE primers (Table 1):
Table 1. Primers used in the SBE reaction for determining the mannose-binding lectin gene (MBL2) single nucleotide polymorphisms (SNPs).
|
Rs number |
Nucleotide substitution 1 (location, amino acid substitution/ SNP position 2) |
Allele |
Primer sequence (5'→ 3') |
DNA strand |
Concentration of the primer in the SBE primer mix (μM) |
|
rs1800450 |
161G>A (exon 1, Gly54Asp) |
A/B |
CCAGGCAAAGATGGGYGTGATG 3 |
forward |
0.1 |
|
rs1800451 |
170G>A (exon 1, Gly57Glu) |
A/C |
TTTTACGTACCTGGTTCCCCCTTTTCT |
reverse |
0.18 |
|
rs7096206 |
-290C>G (promotor, ‑221 2) |
X/Y |
TTTTTTTTGGTCCCATTTGTTCTCACTGCCAC |
forward |
0.06 |
|
rs5030737 |
154C>T (exon 1, Arg52Cys) |
A/D |
TTTTTTTTTTTTCCCTTTTCTYCCTTGGTGYCATCAC 3 |
reverse |
0.08 |
|
rs11003125 |
-619C>G (promotor, ‑550 2) |
H/L |
TTTTTTTTTTTTTTTTGGAGTTTGCTTCCCCTTGGTGTTTTA |
reverse |
0.08 |
|
rs7095891 |
-66C>T (5' UTR of exon 1, +4 2) |
P/Q |
TTTTTTTTCAGGGAAGGTTAATCTCAGTTAATGAACACATATTTACC |
reverse |
0.06 |
1 SNP position according to the database NCBI dbSNP HGVS names NM_000242.2:c entry. 2 SNP position according to the literature [8]. 3 Letter Y in the primer sequence indicates a site containing degenerate base (cytosine or thymine).

Reviewer 2 Report
This manuscript is a very straightforward description of a mannose binding lectin gene variant assay. The paper is clear and well written. I have no issues with the manuscript.
Author Response
Thank you.
Round 2
Reviewer 1 Report
The authors have improved the writing of the text and especially uploaded the final version of the manuscript. However, how much the SNaPshot assay is an innovative diagnostic test is still not clear. None of the required experiments were performed. It would be interesting to see at least one table that clearly shows the aspects of each technique currently used with the same purpose to understand the strengths and weaknesses of each, to convince of the superior utility of the SNaPshot assay compared to the others.
Author Response
Reviewer 1 (Round 2)
Comment: The authors have improved the writing of the text and especially uploaded the final version of the manuscript. However, how much the SNaPshot assay is an innovative diagnostic test is still not clear. None of the required experiments were performed. It would be interesting to see at least one table that clearly shows the aspects of each technique currently used with the same purpose to understand the strengths and weaknesses of each, to convince of the superior utility of the SNaPshot assay compared to the others.
Answer:
We would like to thank the referee for his comment, which revealed that we have not emphasized the benefits of our approach sufficiently in the paper. We do not claim in our paper that SNaPshot assay is an innovative test, the principle is definitely known for some time. We, however, believe that our application and optimization of this method for the purpose of detecting all single nucleotide polymporphisms in MBL2 is indeed innovative as it is the first method capable of identification of all six polymorphisms in a single automated analysis. Previous reports on the use of SNaPshot analysis for detecting MBL2 polymorphisms always focused only on a few of these, never on providing the full analysis of all SNPs. Other methods that can be used for the same purpose either require performing multiple analyses, or cannot be automated.
We have amended the Discussion to better emphasize this and added the table as required. We acknowledge that this correction greatly contributed to better understanding of the pros and cons of our method.
Round 3
Reviewer 1 Report
The authors have addressed my concerns.